# Programmable C:G to G:C genome editing with CRISPR-Cas9-directed base excision repair proteins

Liwei Chen[1], Jung Eun Park[1], Peter Paa[1], Priscilla D. Rajakumar[1], Hong-Ting Prekop[1], Yi Ting Chew [1], Swathi N. Manivannan[1] & Wei Leong Chew [1✉]

Many genetic diseases are caused by single-nucleotide polymorphisms. Base editors can correct these mutations at single-nucleotide resolution, but until recently, only allowed for transition edits, addressing four out of twelve possible DNA base substitutions. Here, we develop a class of C:G to G:C Base Editors to create single-base genomic transversions in human cells. Our C:G to G:C Base Editors consist of a nickase-Cas9 fused to a cytidine deaminase and base excision repair proteins. Characterization of >30 base editor candidates reveal that they predominantly perform C:G to G:C editing (up to 90% purity), with rAPOBEC-nCas9-rXRCC1 being the most efficient (mean 15.4% and up to 37% without selection). C:G to G:C Base Editors target cytidine in WCW, ACC or GCT sequence contexts and within a precise three-nucleotide window of the target protospacer. We further target genes linked to dyslipidemia, hypertrophic cardiomyopathy, and deafness, showing the therapeutic potential of these base editors in interrogating and correcting human genetic diseases.

---

[1] Genome Institute of Singapore, Agency for Science, Technology and Research, Singapore, Singapore. ✉email: chewwl@gis.a-star.edu.sg

Many human genetic diseases are caused by single-nucleotide polymorphisms (SNPs), in which the disease and healthy alleles differ by a single DNA base. Base editors can correct these SNPs by converting the targeted DNA bases into another base in a controllable and efficient fashion. Until recently, base editing technology is only efficient in converting of C:G base pairs to T:A base pairs using cytidine base editors (CBEs) and A:T base pairs to G:C base pairs using adenine base editors (ABEs)[1–3], which together represent half of all known disease-associated SNPs. CBEs and ABEs are also known to effect some C:G to G:C edits as byproducts[4,5]. Indeed, two recent reports capitalized on this observation and demonstrated that fusion of APOBEC-nCas9 to uracil DNA glycosylase (UNG) – which induces abasic sites – results in C:G to G:C editing in mammalian cells[6,7].

In contrast to UNG-mediated base excision initiation[6,7], we look to leverage on the cell's innate base excision repair (BER) pathway, in which DNA polymerase β, DNA ligase III, and XRCC1 are major players[8]. Previous work suggests that removal of uracil glycosylase inhibitor (UGI) from a CBE might lead to an increase in C:G to G:C levels[2,6,7]. UGI inhibits UNG, thereby inhibiting downstream BER[1,2,9]. We speculate that the endogenous BER pathway – in the presence of a bound nCas9 – is linked to the observed C:G to G:C editing and that replacing UGI with BER proteins might increase such editing events. Using a version of CBE – BE3 – as a starting point, we removed UGI, instead fusing rAPOBEC-nCas9 with DNA ligase 3, DNA repair protein XRCC1, or the DNA binding and lyase domain of DNA polymerase β (PB)[10].

In this work, we demonstrate a distinct CGBE architecture that manipulates the BER pathway downstream of abasic site creation. We demonstrate that this class of C:G to G:C Base Editors (CGBEs) edits C:G to G:C (Fig. 1a), which potentially opens up treatment avenues to 11% (singular CGBE) to 40% (CGBE with CBE/ABE) of the disease-associated SNPs in ClinVar (Supplementary Table 1). We further characterize the editing preferences of CGBEs, their activity in clinically relevant cell types, and their off-target profiles.

## Results

**Shortlisting CGBE candidates**. Because the relative orientation of Cas9 fusions may affect the activity of the fusion, we built several versions of our CGBE candidates with the fused rAPO-BEC and BER proteins at different orientations with respect to nCas9 (Fig. 1b). We then separately treated HEK293AAV cells with each candidate along with gRNAs designed to target genomic sites HEK2 and HEK3, both of which were previously used to characterize base editors[1]. As controls, we also treated cells using BE3 and BE4 with the same gRNAs. Since no effective means of C:G to G:C base editing was known at the time, and BE3 has been observed to effect this reaction as a byproduct of C:G to T:A editing[4], we used BE3 as a benchmark for our CGBEs.

High-throughput sequencing of the HEK2 site revealed that our CGBE candidates were able to edit C:G to both G:C and T:A. Out of the 31 candidates tested, 20 candidates showed an increased level of C:G to G:C editing relative to BE3 at position 6 within the protospacer (Supplementary Fig. 1a). The next closest available C was at position 4, where editing levels were lower (<10%, Supplementary Fig. 1b), suggesting a narrower editing window than BE3[1]. In the best performing candidates, up to 29% C:G to G:C editing and 6% C:G to T:A editing were observed for the CGBE candidates, compared with 14 and 36%, respectively, for BE4, and 16 and 18%, respectively, for BE3.

Similarly, sequencing at the HEK3 site revealed editing of C:G to both G:C and T:A. 12 out of 31 candidates gave C:G to G:C editing levels at position 5 that were higher than that achieved by BE3, with the best performing candidates effecting up to 13% C:G

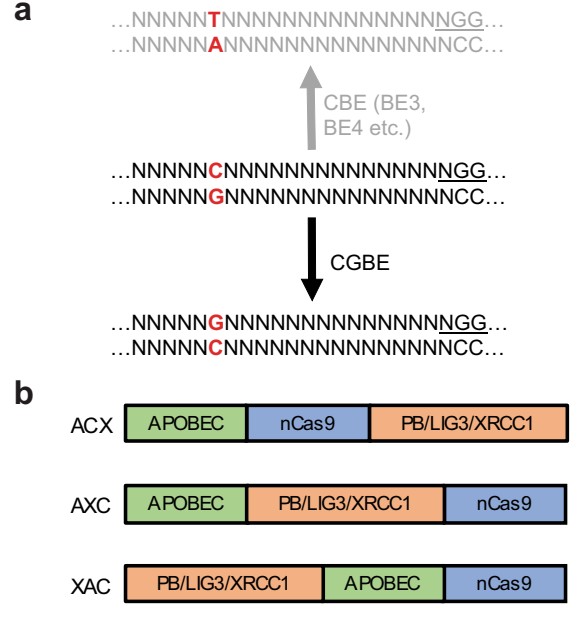

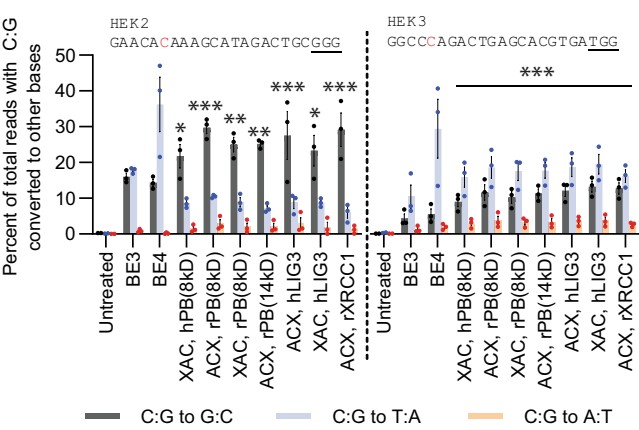

**Fig. 1 Initial screen of CGBE candidates for C:G to G:C editing. a** CBEs like BE3 and BE4 predominantly convert C:G to T:A while CGBE aims to predominantly convert C:G to G:C. **b** CGBE candidates were designed in three orientations – ACX, AXC, and XAC, where X denotes the fused BER protein. **c** Seven candidates were selected for their high C:G to G:C editing at both HEK2 and HEK3. The lower editing at HEK3 is likely due to a disfavored motif (refer to data in Fig. 2a). Targeted C's are in red. PAMs are underlined. *p < 0.05; **p < 0.01; ***p < 0.001 using one-way ANOVA (Dunn- Šidák) of C:G to G:C editing against 'Untreated'. Exact p values are available in Source Data. Each dot represents editing of an individual biological replicate; bars represent mean values; error bars represent SEM of three biologically independent replicates.

to G:C editing compared with 4% for BE3 (Supplementary Fig. 2a). Despite an increase in C:G to T:A editing with CGBEs compared to BE3, a more than twofold increase in C:G to G:C editing with CGBEs led to an increase in the editing ratio of C:G to G:C compared to C:G to T:A. Compared to position 5, C:G to G:C editing at position 3 and position 4 are significantly lower (Supplementary Fig. 2b, c). From this pilot screen, we shortlisted seven of the best performing candidates for a secondary screen (Fig. 1c).

We further tested the CGBEs for C:G to G:C editing at four genomic sites known to be amenable to BE3-mediated editing – EMX1, HEK4, RNF2, and FANCF (Supplementary Fig. 3a). With CGBEs, C:G to G:C edits were efficiently induced (17–24%,

compared to 8–10% with BE3) as the predominant product (up to 69% purity) at *HEK4* and *RNF2*. At sites *EMX1* and *FANCF*, up to 9% C:G to G:C editing was observed despite this not being the predominant edit, while BE3 effected up to 3% C:G to G:C editing. Across the sites tested, CGBE candidates exhibit higher indel rates than BE3 (Supplementary Fig. 4a). We suspect that the generation of an abasic site – which is essential for BER – may have led to higher indel rates. Comparably for recently published UNG-based CGBEs, which leverage abasic site generation, also reported mean indel rates of 10.4%[6]. From this secondary screen, we chose ACX, rXRCC1 and ACX, rPB(8kD) for further characterization.

**Characterizing CGBE editing preferences**. We next determined if target sequence context impacts editing efficiency. In characterizing the two CGBEs with four gRNAs targeting four disease-associated sites – including dyslipidemia-associated gene *ADRB2*[11], hearing loss-associated gene *GJB2*[12], and hypertrophic cardiomyopathy-associated gene *MYBPC3*[13] – we noticed that not only did CGBEs efficiently interrogate disease-associated genes, but they also gave higher levels of C:G to G:C editing at C's immediately following an A/T (Supplementary Fig. 3b, Supplementary Fig. 5, and Supplementary Table 2). Hence, we suspected that the difference in editing efficiency between gRNAs might be due to the different motifs within which the targeted C is located (e.g., ACA at *HEK2*, CCA at *HEK3*; targeted C is underlined). To address such potential differences, we designed 16 gRNAs targeting 16 different *HEK2* sites. These gRNA-site combinations collectively cover all possible NCN motifs with targeted C's at position 6 (Fig. 2a, Supplementary Fig. 3c). Sequencing revealed that the most readily edited motif was ACA, while the least edited motif was GCC. Up to 10% C:G to G:C editing was observed for ACA, ACC, ACT, GCT, TCA, and TCT. This observation applied to both ACX, rPB(8kD) and ACX, rXRCC1. From these results, we propose that the favored DNA motifs are WCW, ACC, and GCT (W is either A or T, Fig. 2b, Supplementary Fig. 6). This understanding is consistent with our observation that C:G to G:C editing is more efficient at *HEK2*, *HEK4*, and *RNF2*, which all had favored DNA motifs of ACA, ACT, and TCT, whereas *EMX1*, *FANCF*, and *HEK3*, with suboptimal DNA motifs of TCC and CCA, are less amenable to C:G to G:C editing. With a preference for WCW, ACC, and GCT, our CGBEs most efficiently target 6 out of 16 possible motifs.

We previously observed a reduction in editing levels at position 4 relative to position 6 of *HEK2* (Supplementary Fig. 1). Similarly, editing levels at position 3 and position 4 of *HEK3* are lower than that at position 5 (Supplementary Fig. 2). Both observations suggested that our candidates might not inherit the exact base editing window of BE3[1]. To elucidate the editing window of the CGBEs, we chose a genomic site that has an alternating 5′-W-C-W-C-W-C-3′ sequence such that gRNAs can be designed with C's located either at every odd position or at every even position (Fig. 2c). We assessed BE3 C:G to T:A editing rates between positions 1 and 9, finding higher editing efficiency between positions 4 and 8. For both ACX, rPB(8kD) and ACX, rXRCC1, C:G to G:C editing was observed in a nine nucleotide window from positions 2 to 10. However, only at positions 5 and 6 was C:G to G:C editing the predominant and appreciable outcome.

Recognizing that simply removing UGI can potentially increase C:G to G:C editing at the expense of C:G to T:A editing[2,6,7,14,15], we next sought to quantify the effect of fusing rXRCC1 or rPB (8kD) to rAPOBEC-nCas9. Across 28 independent treatments using a variety of gRNAs (Fig. 3, Supplementary Fig. 7), BE3 was able to effect on average 4.4% C:G to G:C editing. Removing UGI from BE3 raised mean C:G to G:C editing to 11.5% but increased

undesired indel byproducts to 5.0% (Supplementary Fig. 4b). Fusing rPB(8kD) did not provide additional significant effect ($p > 0.05$). However, fusing rXRCC1 further raised mean C:G to G:C editing levels to 15.4% ($p < 0.01$) and decreased undesired indel rates (3.4%), potentially due to an equilibrium shift from abasic site to repaired base. Meanwhile, the major byproduct C:G to T:A editing stayed between 6 and 9%. Our results indicate that UGI removal from BE3 promotes C:G to G:C editing but increases byproducts; rXRCC1 fusion further increases C:G to G:C editing and decreases byproducts. Therefore, we narrowed down to rAPOBEC-nCas9-rXRCC1 as the preferred CGBE embodiment that effects C:G to G:C editing at $15.4 \pm 7\%$ efficiency in human cells, at a $68 \pm 14\%$ purity, within a three-nucleotide target window and WCW, ACC, and GCT target sequence contexts.

**Off-target activities and activities in other cell types**. As with CRISPR-Cas systems, base editors have been reported to exhibit potential DNA and RNA off-target effects[16–19]. Because CGBEs share the same APOBEC-nCas9 component with BE3, we assessed CGBE and BE3 activities side-by-side on 29 off-target sites using 5 gRNAs. CGBE and BE3 induced >0.1% C:G to D:H edits at the same 15 positions (D is either A, G or T; H is either A, C, or T). Only at 2 out of these 15 positions did CGBE induce greater off-target editing than BE3; at the remaining 13 sites, CGBE showed lower off-target activity. While reduction of off-target activity can be attributed to lower C:G to T:A editing at off-target sites, C:G to G:C editing at the same off-target sites increased (Supplementary Fig. 8). We did not find a site where CGBE induced off-target editing but BE3 did not. These results indicate that CGBE exhibits lower off-target activity at the off-target sites identified for Cas9 and for the BE3 architecture. Nevertheless, there are potentially vast sequence-independent DNA and RNA off-target effects[17–19] that would require a detailed dissection in further work before direct translation of CGBEs.

One limitation of BE3 is its low efficiency in some cell types[20]. We suspect that such limitations might apply to our CGBEs. With BE3, we observed low C:G to T:A editing in H9 stem cells at five genomic sites (with a maximum of 1.2% C:G to T:A editing at *HEK4*; Supplementary Fig. 9). Expectedly, our CGBEs exhibit similarly low C:G to G:C editing efficiencies in the H9 stem cells when evaluated side-by-side. The low editing might be due to chromosomal abnormalities[21] and different methylation profiles in stem cells[22]. Since APOBEC is inefficient at deaminating methylated cytidines[23], further APOBEC engineering[24–26], alongside codon optimization[20,27], might be needed to enhance the efficiency of CGBEs in stem cells. In contrast, in the eHAP cell line, we observed moderate levels of editing with the CGBE (ACX, rXRCC1) inducing up to 8.5% C:G to G:C editing at the *RNF2* and *VEGFA* sites, while BE3 induced 0.9% C:G to T:A editing (Supplementary Fig. 10). The eHAP data suggests that our CGBEs may be moderately efficient in some cell types even if different base editing technology is not. In HTB9 cells – a urinary bladder cancer cell line, both BE3 and our CGBEs can efficiently induce the desired mutations at many sites (up to 17% C:G to G:C editing with CGBE and up to 18% C:G to T:A editing with BE3; Supplementary Fig. 11). ACX, rPB(8kD) appears to consistently outperform ACX, rXRCC1 in HTB9 cells, indicating that different CGBE architecture can be employed for optimal performance according to cell types. We demonstrated that CGBEs is functional across multiple cell types, and that absolute efficiency is partially dependent on the cell type and state, features shared with previous base editors.

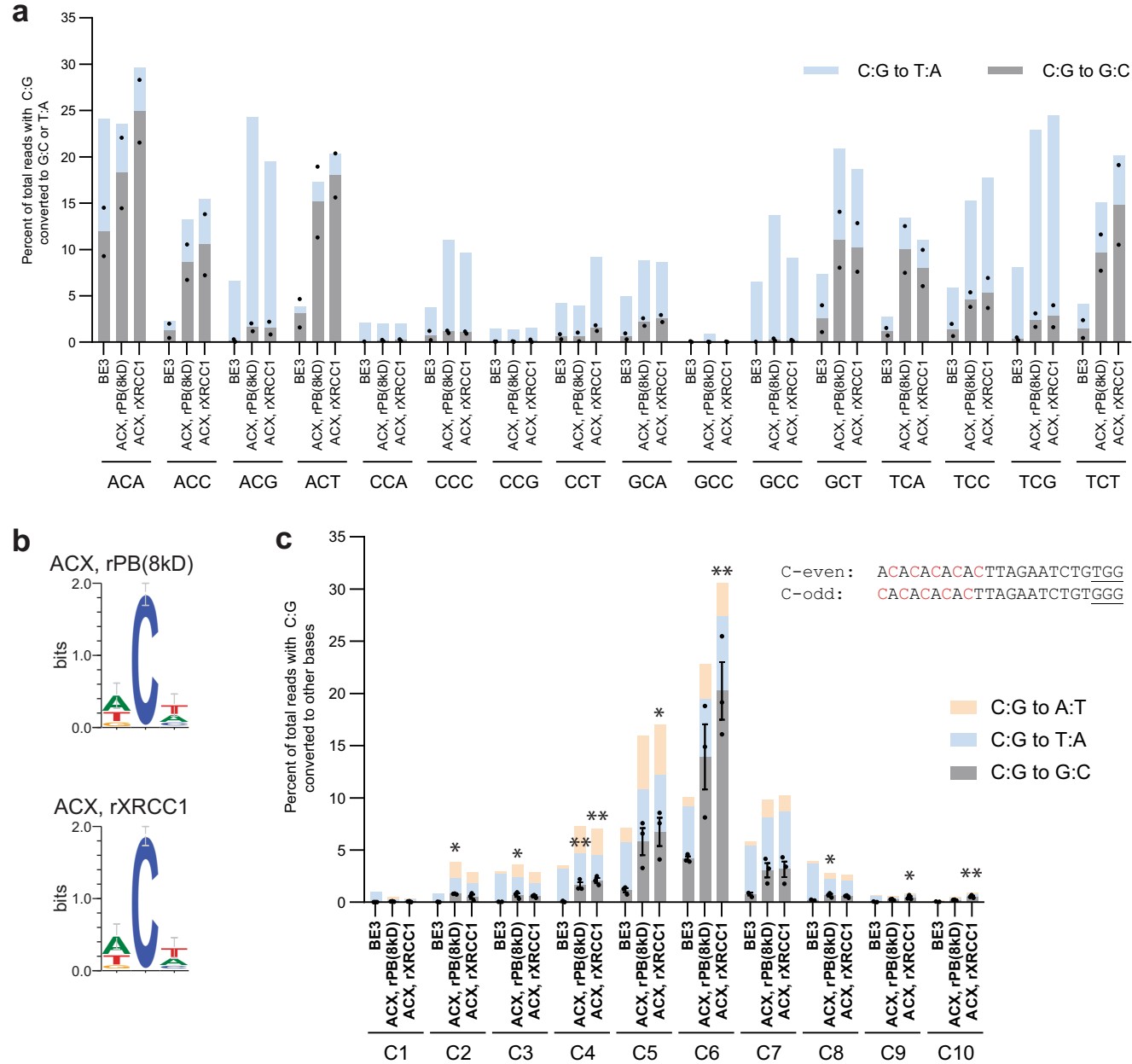

**Fig. 2 Sequence context and editing window for two selected CGBEs. a** Evaluation of C:G to G:C editing at each NCN DNA motif. 16 different gRNAs were designed to target the genomic region around the *HEK2* site (*HEK2-1* to *HEK2-16* in Supplementary Table 2), chosen such that the gRNA-to-target combinations together cover all NCN motif contexts and that genomic distance among gRNAs is minimized (14/16 gRNAs, including the initial *HEK2-1* gRNA:target, reside within a 1.8 kb region, while the other 2 gRNAs target within 10 kb). Each dot represents C:G to G:C editing of an individual biological replicate; bars represent mean values of two biologically independent replicates. **b** DNA WebLogo created with target motifs in which C:G at position 6 was edited to G:C ($n = 2$ biologically independent replicates; error bars are Bayesian 95% confidence intervals). **c** Editing window of CGBEs using gRNAs with alternating 5'-W-C-3' motifs. Targeted C's are in red. PAMs are underlined. *$p < 0.05$; **$p < 0.01$ using one-way ANOVA (Dunn-Šidák) of C:G to G:C editing against 'BE3'. Exact $p$ values are available in Source Data. Each dot represents C:G to G:C editing of an individual biological replicate; bars represent mean values; error bars represent SEM of three biologically independent replicates.

## Discussion

Here, we developed CGBEs that target cytidine within a specified window and convert it into guanine as a predominant editing product. In separate works, Liu and Koblan designed CGBE candidates by fusing UDG (UNG), UdgX, and polymerases with rAPOBEC-nCas9[28] (Supplementary Fig. 12); Zhao et al. and Kurt et al. developed CGBEs fusing UNG to rAPOBEC-nCas9[6,7]. All four CGBE studies induce a C to U change via rAPOBEC and envision this U to be further converted to an abasic site; but the strategy involved and downstream resolution of this abasic site

are distinctly different. Zhao et al. and Kurt et al. hypothesized that fusing UNG to rAPOBEC-nCas9 increases the amount of C:G to G:C edits by further increasing the generation of abasic sites (Supplementary Fig. 13). In contrast, we focused on manipulating DNA repair downstream of abasic site creation, reasoning that APOBEC-nCas9 is already able to generate a significant amount of C:G to G:C editing (Fig. 3a, BE3 (no UGI)). Under similar reaction conditions used in Kurt et al., UNG-based CGBEs reported similar mean C:G to G:C editing in unselected cells (14.4% vs. our 15.4%)[6]. The distinct approaches suggest the

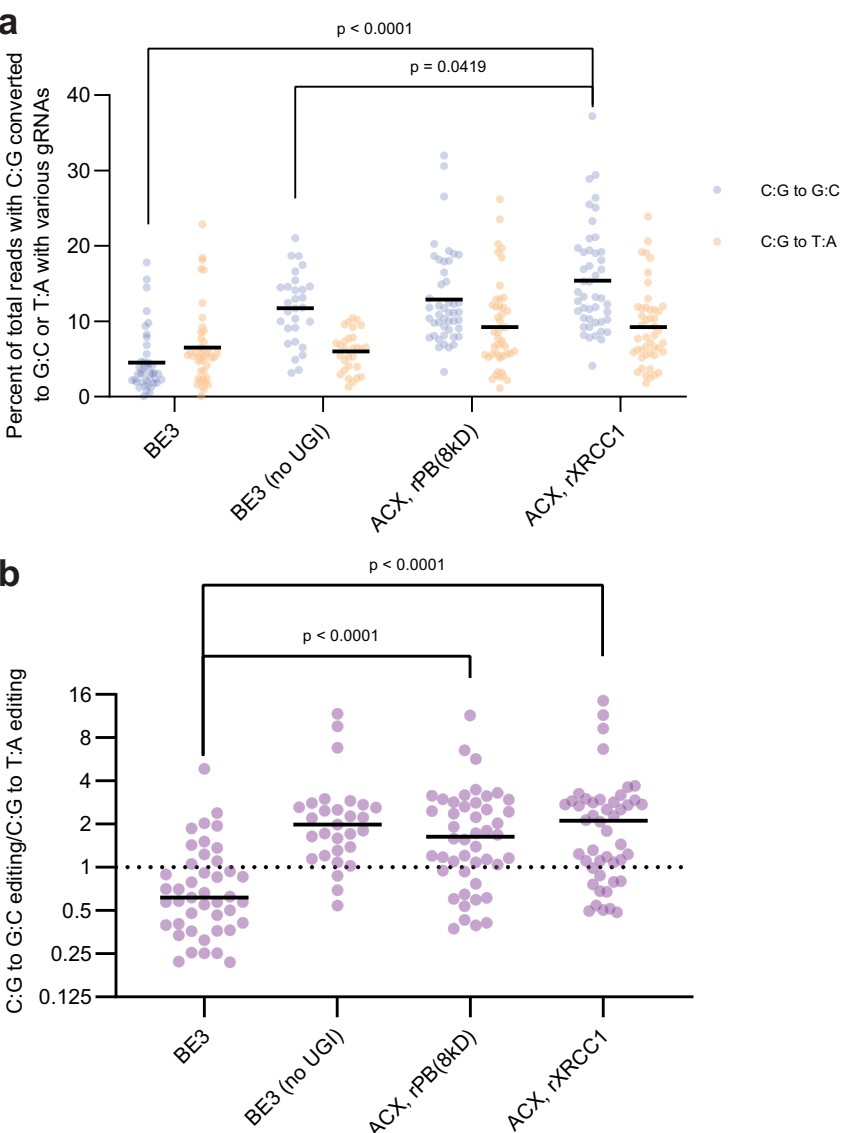

**Fig. 3 CGBE induces efficient C:G to G:C editing as the predominant product. a** Removal of UGI from BE3 increases C:G to G:C editing (blue); fusion of rXRCC1 further increases C:G to G:C editing. The major byproduct is C:G to T:A editing (orange). **b** Mean C:G to G:C editing/C:G to T:A editing ratio. For both plots, *p* values were obtained via Mann–Whitney tests between the indicated editors. Each dot represents editing of an individual biological replicate. gRNAs chosen have C's within WCW, ACC, or GCT motifs. Black lines represent mean values of 43 (BE3), 29 (BE3 (no UGI)), or 46 (ACX, rPB(8KD) and ACX, rXRCC1) biologically independent replicates.

possibility that a combination of an abasic site creation strategy and a BER-manipulating strategy may lead to the next iterations of enhanced CGBEs.

Koblan and Liu's work seeks to induce base excision and perform a translesion polymerization across the targeted abasic site (Supplementary Fig. 13). A translesion polymerization hypothesis was also put forth by Gajula[29]. This proposed mechanism is unlikely to be the case for our CGBEs. Instead, we envisage that upon creation of an abasic site in the Cas9-induced R-loop, cellular UNG is displaced by APE1, after which XRCC1 recruits various BER components[8,30] to repair the abasic site independently of the unedited opposite strand, giving rise to guanine as the major product. Subsequent DNA repair converts the G:G mismatch to G:C. Such a hypothesis would be consistent with (a) the tight binding of Cas9 on its target strand[31], which renders that strand less accessible to other enzymes; (b) the accessibility of the deaminated strand as a single-strand of the R-loop[32,33]; (c) the detrimental effect of UDG and UdgX on C:G to

G:C editing[28] (Supplementary Fig. 12), which suggests that the persistence of an UDG-bound site or abasic site might act against the C:G to G:C reaction; and (d) the C:G to G:C editing effected by our CGBEs with domains that do not have intrinsic polymerase activity but are key drivers of abasic site repair. While further mechanistic studies and development continue, CGBEs expand the growing suite of precise genome-editing tools that include CBEs[1,2], ABEs[3], CGBEs[6,7,28], and prime editors[32] (Supplementary Fig. 12), together enabling the precise and efficient engineering of DNA for research, biological interrogation, and disease correction.

## Methods

**Constructs and molecular cloning**. The BE3 (Addgene plasmid #73021), prime editor 2 (Addgene plasmid #132775), pegRNA-HEK3_CTT_ins (Addgene plasmid #132778) plasmids used in our study are gifts from David R. Liu. The BE3 plasmid is a mammalian expression plasmid with BE3 being driven by a CMV promoter. hXRCC1 (pTXG-hXRCC1) and hLIG3 (pGEX4T-hLIG3) are gifts from Primo Schaer (Addgene plasmid # 52283 and # 81055, respectively). The mutation R400Q is

introduced to hXRCC1 and N628K is introduced to hLIG3 via blunt-end ligation. Briefly, plasmid containing either hXRCC1 or hLIG3 is amplified via PCR using Q5 Hot Start HiFi 2X Master Mix (NEB, M0494). PCR product is then treated with DpnI (NEB, R0176) and T4 Polynucleotide Kinase (NEB, M0201) at 37 °C for 30 min and inactivated at 65 °C for 20 min before being ligated using T4 DNA Ligase (NEB, M0202; room temperature for 2 h). Ligated product is then transformed into chemically competent 5-alpha E. Coli (NEB, C2987). rXRCC1, rLIG3, hPBs, and rPBs were obtained as human codon-optimized de novo synthesized gene fragments (Twist Biosciences). All other oligonucleotides used in the study were de novo synthesized (IDT DNA). To fuse BER proteins with rAPOBEC-nCas9, Q5 Hot Start HiFi 2X Master Mix was used to generate Gibson fragments of the BER proteins as Gibson inserts. After checking PCR products on a gel, Gibson insert and vector were incubated with NEBuilder HiFi DNA Assembly Master Mix (NEB, E2621) for 1 h at 50 °C. The Gibson reaction is then transformed into chemically competent E. Coli.

All assembled plasmids were Sanger sequenced for sequence verification and were prepared using either the PureYield Plasmid Miniprep System (Promega, A1223) or Plasmid Plus Maxi Kit (Qiagen, 12965).

**Cell culture.** HEK293AAV cells (Agilent, 240073) were maintained in DMEM with GlutaMAX and sodium pyruvate (Thermo Fisher, 10569-010) supplemented with 10% HI FBS (Thermo Fisher) at 37 °C and 5% $CO_2$. HTB9 cells (ATCC, 5637) were maintained in RPMI-1640 with L-glutamine and sodium bicarbonate (Sigma, R8758) supplemented with 10% HI FBS (Thermo Fisher) and 1% MEM non-essential amino acids solution (Thermo Fisher, 11140050) at 37 °C and 5% $CO_2$. Both HTB9 and HEK293AAV cells were transfected via lipofection. After cells reached ~80% confluency, they were washed with PBS, pH 7.2 (Thermo Fisher, 20012-027) before being treated with TrypLE Express (Thermo Fisher, 12604). 30,000 cells were added to each well of a 48-well plate one day before transfection. For each well, 750 ng of base-editor plasmid, 250 ng of gRNA plasmid, and 20 ng of GFP plasmid were transfected into these cells using Lipofectamine 3000 (Invitrogen, L3000015) according to the manufacturer's protocol. The media were replaced with fresh media 24 h after transfection. No selection was done on the cells. For prime editing, 750 ng of PE plasmid, 250 ng of pegRNA, and 83 ng of sgRNA were used for transfection. 72 h after lipofection, media were removed; cells were washed with 50 μL PBS, pH 7.2, and genomic DNA was extracted using 50 μL of Quick Extract DNA Extract Solution (Lucigen, QE09050) per well according to manufacturer's protocol. All sample sizes indicate biological replicates.

eHAP cells (Horizon Discovery, C669) were maintained in IMDM (Thermo Fisher, 31980-030) supplemented with 10% FBS at 37 °C and 5% $CO_2$. 200,000 cells were nucleofected with 750 ng of base editor and 250 ng of gRNA expression plasmids using the SE Cell Line 4D-Nucleofector X Kit S (Lonza) and program DS-138 on the 4D X-Unit. Cells were harvested 72 h after nucleofection without any selection.

H9 stem cells (WiCell, WA09) were maintained in mTeSR1 (Stemcell technology, 85850). 200,000 cells were nucleofected with 1500 ng of base editor and 500 ng of gRNA expression plasmids using the P3 Primary Cell kit (Lonza, V4XP-3024) and program hES H9 program on the 4D X-Unit. Cells were harvested 72 h after nucleofection without any selection.

**Sequencing of genomic DNA.** Sites of interest were prepared for high-throughput sequencing via two PCR amplifications – the first PCR amplifies the region of interest while the second PCR adds appropriate sequencing barcodes. Briefly, after Quick Extract treatment, genomic DNA from 1 uL of cell lysate was amplified in a 25 uL reaction using Q5 DNA polymerase (NEB, M0491). Primers for PCR1 are listed in Supplementary Table 3. Without further purification, 1 uL of PCR1 reaction mixture was used for PCR2 (25 uL reaction). Primers for the PCR2 are based off Illumina adaptors. Amplicons from the PCR2 were then pooled and gel extracted (Promega, A9282) to make the final library, which was quantified via Qubit fluorometer (Thermo Fisher) and sequenced on an Illumina iSeq 100 according to the manufacturer's protocol. The resultant FASTQ files were analyzed using CRISPResso2[34]. All sample sizes indicate biological replicates.

Statistical analyses were performed on MATLAB R2016a, Microsoft Excel 2016, and GraphPad Prism 9. Weblogos were created using Weblogo 3[35].

**Reporting summary.** Further information on research design is available in the Nature Research Reporting Summary linked to this article.

## Data availability

Sequences of ACX, rXRCC1, ACX, rPB(8kD), other BER proteins, and pegRNAs are available in the Supplementary Information. High-throughput sequencing data can be accessed via NCBI Sequence Read Archive database with SRA accession code PRJNA692655 and BioProject accession code PRJNA692655. Plasmids encoding ACX, rPB(8kD) (Addgene plasmid # 165445) and ACX, rXRCC1 (Addgene plasmid # 165444) are available on Addgene. Source data are provided with this paper.

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

## Acknowledgements

We thank Anna Traczyk for helpful discussions; Eddie Keng for help with HTS; Hannah Nicholas, Nicholas Ong for editing; and Ke Guo and Daryl Lim for general administrative support. This work is supported by Agency for Science, Technology and Research (A*STAR) and Industrial Alignment Fund Pre-Positioning (IAF-PP) grant H17/01/a0/ 012. Work by P.D.R. and W.L.C. is further supported by Advanced Manufacturing and Engineering (AME) Programmatic grant A18A9b0060.

## Author contributions

L.C. made the initial discovery. L.C., J.E.P., and W.L.C. designed the research. L.C., J.E.P., and P.P. performed the experiments. P.D.R. performed biological replicates. Y.T.C. and S.N.M. cloned plasmids. L.C., J.E.P., P.P., H.-T. P., and W.L.C. analyzed data. L.C., H.-T. P., and W.L.C. wrote the manuscript with input from other authors. W.L.C. supervised the research.

## Competing interests

L.C. and W.L.C. are named inventors on a patent application based on this work. Patent applicant: Agency for Science, Technology and Research. Application number: 10201913340Q. Status of Application: PCT submitted; Specific aspect of manuscript covered in patent application: design, applications, and uses of base editors with base excision repair proteins (Figs. 1a, 1b, 3, SI Fig. 3, 5, 7, 9, 10, 11, 13) although all data presented here are included in patent application. The remaining authors declare no competing interests.
