## [Peer Review File · Nature Communications]

Reviewers' Comments:

Reviewer #1:

Remarks to the Author:

The revised manuscript from Chen et al describes C-to-G base editors (CGBEs). I refereed the original manuscript, and the "Point by point response" below is formatted to adhere to the numbering of the authors' responses to my original concerns.

Overall, the revised manuscript contains some important improvements over the initial version. The authors have partly addressed previous concerns with additional analyses and provide clarification or more accurate formulations in other aspects. These include analyses of indel-frequencies at on-target sites, profiling of some sequence-specific off-target editing and acknowledgement of some remaining gaps.

Since the initial submission, two studies have reported other CGBEs. These approaches have with similar limitations in product purity and on-target efficacy to those described in the current manuscript (Kurt et al. 2020, Zhao et al. 2020). The CGBE candidates of Chen et al. present a completely orthogonal approach that could therefore aid the design of combined BE architectures in future. Therefore, I would support the publication of Chen et al.'s manuscript in *Nature Communications*, if the following remaining concerns are supported by additional data or appropriate changes to the manuscript.

Point-by-point response:

[3-1] Low product purity and modest activity.

Regarding product purity, the additional data on indel-formation indeed suggests a modest improvement of CGBE over BE3-no-UGI. Also recently published alternative CGBEs seem to struggle with high C>T editing at some sites. Interestingly, sites with predominant C>T editing are not identical between the different CGBE strategies (e.g. FANCF in Extended Data Fig. 3 compared to FANCF site 1 in Kurt et al). If the authors wish, it could be worth highlighting these differences to existing CGBE candidates.

Regarding editing efficiency, I disagree with the author's response in that BE3 and BE3-no-UGI are benchmarks for G>C editing, since they were not designed for this purpose. Nonetheless, the newly included comparison of Cas9-BER fusions with recently published CGBEs or prime editors suggest their merit as orthogonal CGBE strategy. I agree with the more careful statements on immediate applications of their technology and with the potential of combining rAPOBEC-nCas9-BER with other CGBE-strategies in future studies. I view this concern as addressed.

[3-2] Sequence-specific and non-specific Off-target activity on DNA and RNA.

The additional analyses in Extended Data Figure 8 represents a first step to assessing sequence-dependent off-target activities of rAPOBEC-nCas9-BER CGBEs. However, the corresponding paragraph (lines 131-141) neither directly mentions the vast sequence-independent DNA and RNA off-target effects reported for other APOBEC-containing BEs, nor does it provide any data to rule it out. Without further changes, the paragraph is misleading to parts of the broad audience addressed by *Nature Communications* with respect of the maturity and potential for direct translation of the technology.

[3-3] Cell/tissue context dependency of rAPOBEC-nCas9-BER activity

Context-specific activity was neither addressed experimentally nor mentioned in the revised

manuscript, although this concern has been brought up by all three referees. This point cannot be entirely omitted, given the well-documented variability of editing efficiencies across tissue- /cell types and given that HEK293-derived cells represent a notoriously low bar for transfection-based editing tools.

[3-4] Introduction of indels by rAPOBEC-Cas9-BER
Extended Data Fig. 4 addresses my concerns.

Minor concerns

[3-5] Speculation on mechanisms

I thank the authors for the much clearer formulation and schematics. I agree that, especially in light of other proposed CGBE strategies, this information is important to include.

[3-6] "opens up avenues to treat half of all known SNP-associated human genetic diseases"

I thank the authors for the overall more accurate wording. This particular sentence still refers to 'SNP-associated diseases', while Extended Data Table 1, to the best of my knowledge, reports percentages of 'disease associated SNPs'. These two measures are not identical.

[3-7] Description of the exact reagents used.

This point is fully addressed by the additional clarification in the manuscript and the supplemental information.

Reviewer #2:

Remarks to the Author:

The authors have included some new data in the revised manuscript, in response to concerns about on-target/off-target activity of CGBEs, and have included a comparison to prime editing.

I appreciate the extra experimental work and analysis, but the authors have ignored a major concern raised in the first iteration of the manuscript - the validation of the efficiency of these tools in 'real-world' contexts such as non-HEK293, disease-relevant cell types or model systems where such tools would be used. They argue that "...that efficiency of CGBE in C>G is comparable to the state-of-the-art of BE3 in C>T, using the same conditions and assays." However, it is important to remember that the original BE3 tools were in fact quite inefficient in non-HEK293 systems. Hence, now it is even more important to provide some level of confidence that new tools can function in research-relevant settings.

In the original version, I raised concerns about the heterogeneity in editing outcomes, meaning that target Cs were often still converted to Ts or As, instead of Gs. In the rebuttal, the authors state that: "CGBE does enable C>G as the most frequent outcome", citing Figure 3, however, most other examples shown in the main figures do not support this (Fig 1c, HEK3; Fig 2a, 9/16 have more T/A than G; Fig 2c, only C6 has more C>G than other outcomes).

I feel that both of the concerns above can be addressed by demonstration of effective C>G editing in a relevant cell type (or two), where C>G is the necessary outcome (not T or A). Without that, I don't believe the authors have provided the data to back up the title of the paper: "Precise and programmable C:G to G:C base editing in genomic DNA"

We thank the reviewers for the insightful comments that have strengthened the manuscript. We provide a point-by-point response below:

Reviewer #2 (Remarks to the Author):

The revised manuscript from Chen et al describes C-to-G base editors (CGBEs). I refereed the original manuscript, and the “Point by point response” below is formatted to adhere to the numbering of the authors’ responses to my original concerns.

Overall, the revised manuscript contains some important improvements over the initial version. The authors have partly addressed previous concerns with additional analyses and provide clarification or more accurate formulations in other aspects. These include analyses of indel-frequencies at on-target sites, profiling of some sequence-specific off-target editing and acknowledgement of some remaining gaps.

Since the initial submission, two studies have reported other CGBEs. These approaches have with similar limitations in product purity and on-target efficacy to those described in the current manuscript (Kurt et al. 2020, Zhao et al. 2020). The CGBE candidates of Chen et al. present a completely orthogonal approach that could therefore aid the design of combined BE architectures in future. Therefore, I would support the publication of Chen et al.'s manuscript in Nature Communications, if the following remaining concerns are supported by additional data or appropriate changes to the manuscript.

Point-by-point response:

[3-1] Low product purity and modest activity.

Regarding product purity, the additional data on indel-formation indeed suggests a modest improvement of CGBE over BE3-no-UGI. Also recently published alternative CGBEs seem to struggle with high C>T editing at some sites. Interestingly, sites with predominant C>T editing are not identical between the different CGBE strategies (e.g. FANCF in Extended Data Fig. 3 compared to FANCF site 1 in Kurt et al). If the authors wish, it could be worth highlighting these differences to existing CGBE candidates.

We thank the reviewer for the insight. While we have shown that the editing profiles (desired C>G editing or C>T byproducts with CGBEs) for these orthogonal base editors differ, particularly because of the different architecture, we believe that further fine-resolution comparison among various base editors would be more appropriately investigated and highlighted in a detailed bioinformatics study.

Regarding editing efficiency, I disagree with the author's response in that BE3 and BE3-no-UGI are benchmarks for G>C editing, since they were not designed for this purpose. Nonetheless, the newly included comparison of Cas9-BER fusions with recently published CGBEs or prime editors suggest their merit as orthogonal CGBE strategy. I agree with the more careful statements on immediate applications of their technology and with the potential of combining rAPOBEC-nCas9-BER with other CGBE-strategies in future studies. I view this concern as addressed.

[3-2] Sequence-specific and non-specific Off-target activity on DNA and RNA.

The additional analyses in Extended Data Figure 8 represents a first step to assessing sequence-dependent off-target activities of rAPOBEC-nCas9-BER CGBEs. However, the corresponding paragraph (lines 131-141) neither directly mentions the vast sequence-independent DNA and RNA off-target effects reported for other APOBEC-containing BEs, nor does it provide any data to rule it out. Without further changes, the paragraph is misleading to parts of the broad audience addressed by Nature Communications with respect of the maturity and potential for direct translation of the technology.

We have now added in the Main Text in Lines 139 to 141 that “Nevertheless, there are potentially vast sequence-independent DNA and RNA off-target effects (17-19) that would require a detailed dissection in further work before direct translation of CGBEs” so as to make clear that our off-target analysis a first step in evaluating CGBE specificity.

[3-3] Cell/tissue context dependency of rAPOBEC-nCas9-BER activity

Context-specific activity was neither addressed experimentally nor mentioned in the revised manuscript, although this concern has been brought up by all three referees. This point cannot be entirely omitted, given the well-documented variability of editing efficiencies across tissue-/cell types and given that HEK293-derived cells represent a notoriously low bar for transfection-based editing tools.

We have now evaluated our CGBEs in multiple cell lines, namely the cancer cell line HTB9, the well-known haploid human cell eHAP, and the H9 stem cells. Indeed as the reviewer pointed out, there is variability of editing efficiency across cell types. We have included the detailed data and text in Lines 142 to 161.

[3-4] Introduction of indels by rAPOBEC-Cas9-BER
Extended Data Fig. 4 addresses my concerns.

Minor concerns

[3-5] Speculation on mechanisms

I thank the authors for the much clearer formulation and schematics. I agree that, especially in light of other proposed CGBE strategies, this information is important to include.

[3-6] “opens up avenues to treat half of all known SNP-associated human genetic diseases”

I thank the authors for the overall more accurate wording. This particular sentence still refers to 'SNP-associated diseases', while Extended Data Table 1, to the best of my knowledge, reports percentages of 'disease associated SNPs'. These two measures are not identical.

This line has been revised. It now reads “which potentially opens up treatment avenues to 11% (singular CGBE) to 40% (CGBE with CBE/ABE) of the disease-associated SNPs in ClinVar.”

[3-7] Description of the exact reagents used.

This point is fully addressed by the additional clarification in the manuscript and the supplemental information.

Reviewer #3 (Remarks to the Author):

The authors have included some new data in the revised manuscript, in response to concerns about on-target/off-target activity of CGBEs, and have included a comparison to prime editing.

I appreciate the extra experimental work and analysis, but the authors have ignored a major concern raised in the first iteration of the manuscript - the validation of the efficiency of these tools in 'real-world' contexts such as non-HEK293, disease-relevant cell types or model systems where such tools would be used. They argue that "...that efficiency of CGBE in C>G is comparable to the state-of-the-art of BE3 in C>T, using the same conditions and assays." However, it is important to remember that the original BE3 tools were in fact quite inefficient in non-HEK293 systems. Hence, now it is even more important to provide some level of confidence that new tools can function in research-relevant settings.

We thank the reviewer for the insight and refer to the response in point [3-3] above and the data across 3 additional cell types now included.

In the original version, I raised concerns about the heterogeneity in editing outcomes, meaning that target Cs were often still converted to Ts or As, instead of Gs. In the rebuttal, the authors state that: "CGBE does enable C>G as the most frequent outcome", citing Figure 3, however, most other examples shown in the main figures do not support this (Fig 1c, HEK3; Fig 2a, 9/16 have more T/A than G; Fig 2c, only C6 has more C>G than other outcomes).

The reviewer points out cases in which the targeted C is not in the most favorable context (either not C6 and/or not located within a favored editing motif as determined by our results). To ensure clarity, we stated in the Figures 1 and 2 legends that these target C's reside within disfavored motifs. Secondly, Figure 3 is a more unbiased depiction because it includes all sites tested in the HEK293 cells, whereas individual figure panels depict individual sites or results from the intermediate development process prior to the final CGBE design. We hope that the manuscript is now clear that within this sequence context and across most sites, CGBE enables C>G as the most frequent outcome.

I feel that both of the concerns above can be addressed by demonstration of effective C>G editing in a relevant cell type (or two), where C>G is the necessary outcome (not T or A). Without that, I don't believe the authors have provided the data to back up the title of the paper: "Precise and programmable C:G to G:C base editing in genomic DNA"

We have now included new data with CGBEs in three additional cell types. We have also revised the title to more accurately describe the paper.

Reviewers' Comments:

Reviewer #1:

Remarks to the Author:

The authors' changes have addressed the minimum of my comments on the previous revision. This satisfies my former concerns, though just barely.

Reviewer #2:

Remarks to the Author:

The authors have adequately addressed my concerns raised in the previous review.